# Calanus Oil and Lifestyle Interventions Improve Glucose Homeostasis in Obese Subjects with Insulin Resistance

**DOI:** 10.3390/md23040139

**Published:** 2025-03-23

**Authors:** Felix Kerlikowsky, Madeline Bartsch, Wiebke Jonas, Andreas Hahn, Jan Philipp Schuchardt

**Affiliations:** 1Institute of Food Science and Human Nutrition, Leibniz University Hannover, Am Kleinen Felde 30, 30159 Hannover, Germany; bartsch@foh.uni-hannover.de (M.B.); hahn@foh.uni-hannover.de (A.H.); schuchardt@foh.uni-hannover.de (J.P.S.); 2NutritionLab, Faculty of Agricultural Sciences and Landscape Architecture, Osnabrueck University of Applied Sciences, 49090 Osnabrueck, Germany; 3Institute for Medical Microbiology and Hospital Epidemiology, Hannover Medical School, 30625 Hannover, Germany

**Keywords:** calanus firmarchicus, homeostatic model assessment of insulin resistance, dietary supplements, lifestyle intervention

## Abstract

Obesity increases the risk for insulin resistance (IR) and type-2 diabetes. Lifestyle interventions (LI) and dietary supplementation can help mitigate IR. We investigated the effect of calanus oil (CO) supplementation, combined with LI, on glucose homeostasis in obese subjects. CO, a novel marine oil, contains omega-3 fatty acid wax esters as well as plant sterols and astaxanthin. In the double-blind, randomized, placebo-controlled 12-week intervention study, 266 subjects with distinct IR phenotypes were assigned to four groups: 2 g CO/day, 4 g CO/day, 2 g CO/day + LI, and placebo. The effect of CO on HOMA index reduction was influenced by the initial (t_0_) squared HOMA index (interaction *p* = 0.011). A post hoc test showed significant improvement with 2 g CO/day + LI (estimated marginal means [EMM] 95% confidence interval [CI]: −0.19 [−0.80–0.41], *p* = 0.021). Secondary analysis revealed that 4 g CO/day had significant effects in subjects with mild IR (HOMA index 2.5–5.0) (EMM [95% CI]: −0.76 [−1.53–0.03], *p* = 0.043). CO supplementation improved glucose homeostasis, with effects varying by dose, combination with LI, and IR phenotype.

## 1. Introduction

Insulin resistance (IR) is a metabolic disorder characterized by reduced insulin action in target tissues, leading to impaired glucose homeostasis and type 2 diabetes mellitus (T2DM) [1]. Obesity as a risk factor for IR is already considered a pandemic in western countries and is predicted to affect 24% of the world’s population by 2025 [2]. Importantly, the accumulation of visceral adipose tissue (VAT) plays a key role in the development of IR. The hypertrophy of visceral and ectopic adipocytes induces chronic inflammation that impair peripheral insulin action [3]. Additionally, VAT serves as a marker for ectopic lipid accumulation, leading to the formation of lipotoxic lipids like diacylglycerols and ceramides, which inhibit insulin signaling [4].

The Homeostatic Model Assessment (HOMA) index is a mathematical model, first proposed by Matthews et al., that provides a tool for estimating IR based on fasting blood glucose and insulin levels [5]. It allows the assessment of intrinsic beta cell function and insulin sensitivity and has been validated against the gold standard method for assessing IR (hyperinsulinemic euglycemic clamp), which can only be used in small studies [6]. The HOMA index has been widely used to describe IR, but the thresholds that identify individuals at risk of developing T2DM may vary depending on population, ethnicity, and health status [7,8]. In the Jackson Heart Study, the HOMA index was used as a marker of IR in an obese population, offering additional stratification of T2DM risk beyond obesity alone [9]. However, despite these insights, it remains unclear how to effectively address impaired glucose homeostasis in obese individuals, as energy-restricted diets often fail to produce sustained long-term weight loss, primarily due to low compliance rates [10].

Lifestyle interventions (LI), including dietary restrictions and increased physical activity, alongside pharmacotherapy, can result in clinically significant weight loss and enhanced insulin sensitivity [11]. Besides this basic therapy for obesity and T2DM management, dietary supplements with specific nutrients and bioactive compounds can contribute to overall metabolic and physiological functions. Notably, n3 PUFAs play a crucial role due to their anti-inflammatory properties and their ability to regulate glucose metabolism, lipid profiles, and insulin sensitivity, making them particularly beneficial for conditions such as metabolic syndrome or insulin resistance. However, it should be clearly stated that dietary supplements do not replace a balanced diet in managing obesity.

Calanus oil (CO), derived from the marine crustacean *Calanus finmarchicus*, differs from conventional n3 PUFA-rich oils in the binding form of its fatty acids (FAs). In CO, more than 80% of FAs are bound as wax esters, whereas in fish oil, they are primarily bound to triacylglycerols, and in krill oil, to phospholipids [12]. Furthermore, CO contains antioxidants such as astaxanthin, plant sterols, and fatty alcohols, which may offer additional health benefits [13,14]. In a preclinical study, Höper et al. [15] investigated the effects of supplemental CO on diet-induced obesity in mice and found that CO reduced insulin levels and improved glucose tolerance in mice fed a high-fat diet. In a first-in-human study, preliminary evidence was found that supplementation with 2 g of CO/day improved the parameters of glucose impairment in an obese population with pre-existing IR [16]. However, knowledge of the effects of CO on glucose metabolism is still limited. We hypothesized that CO supplementation, both alone or combined with LI, could improve IR in subjects with abdominal obesity. Therefore, we aimed to examine the effects of 12 weeks of CO supplementation at varying doses, with or without LI, on parameters of glucose metabolism, focusing specifically on the HOMA index in obese subjects with IR. As an exploratory investigation, this study also seeks to identify IR phenotypes most responsive to CO-based preventive strategies (alone or in combination with LI). The findings will help to develop follow-up hypotheses for future research on tailored CO interventions.

## 2. Results

Of the 266 randomized subjects, 29 subjects did not complete the study according to protocol, leaving 237 subjects with data available for statistical analysis (Figure 1).

### 2.1. Study Cohort Characteristics at Baseline

The study cohort was predominantly female (≥66%), aged 30–75 years, and characterized by obesity with a mean BMI of 34.6 ± 5.3 kg/m^2^. No differences in sex, anthropometric variables, or MetS scores between the groups existed before the intervention (t_0_) (Table 1, Appendix A: Study characteristics at baseline (*n* = 266) (study cohort as intention to treat)).

### 2.2. Effects of Intervention on HOMA Index (Primary Outcome)

Before the intervention (t_0_), there were no significant differences in the parameters of glucose impairment between the groups (Table 2). Figure 2 visualizes the effect of 12 weeks of CO supplementation on the HOMA index compared to the placebo across the entire range of the HOMA index at t_0_. The effect was attenuated for low and high HOMA indices at t_0_, implying a significant squared moderating effect of HOMA index levels at t_0_ (group^x^HOMA index t_0_^2^ with *p* = 0.011, Table 2). Post hoc testing showed that the interaction (group^x^HOMA index t_0_^2^) was significant only for 2 g CO/day + LI compared to the placebo (estimated marginal means [EMM] 95% confidence interval [CI]: −0.19 [−0.80–0.41], *p* = 0.021).

A simple slope analysis was conducted, including main group effects, linear, and squared interactions with the HOMA index at t_0_. The slopes represent the estimated effects of CO in the groups of 2 g CO/day; 4 g CO/day, and 2 g CO/day + LI compared to the placebo for different HOMA indices at t_0_. The model was controlled for age, BMI, and sex. Vertical lines delineate the HOMA index at t_0_ ranges for IR phenotypes: I (unlikely), II (mild), and III (severe). Squared interaction (group^x^HOMA index t_0_^2^) was significant with *p* = 0.011. Post hoc analysis revealed that the interaction (group^x^HOMA index t_0_^2^) was only significant in the 2 g CO/day + LI group compared to the placebo group (*p* = 0.021). The abbreviations are CO, calanus oil; IR insulin resistance; and LI, lifestyle intervention.

As an alternative approach, the effects of CO on the HOMA index were examined at each level of the IR phenotype (Figure 3). ANCOVA showed that the interaction of the group^x^IR phenotype was significant (*p* = 0.036). This means that CO had a different effect on study subjects depending on their IR type. Post hoc testing revealed that the significant effect on the HOMA index was among subjects with mild IR (Figure 3b), with a difference in the change of the HOMA index between the 4 g CO/day group vs. the placebo group (EMM [95% CI]: −0.76 [−1.53–0.03], *p* = 0.043) and between the 2 g CO/day + LI group vs. the placebo group (EMM [95% CI]: −0.76 [−1.51–0.02], *p* = 0.032). The group with 2 g CO/day alone had no effect on the HOMA index compared to the placebo in subjects with mild IR. No significant effects of the intervention were observed in subjects with unlikely IR (Figure 3a) or in subjects with severe IR (Figure 3c).

### 2.3. Secondary Outcomes

Similar to the results observed for the HOMA index (Figure 2), we found that the effect of CO on the secondary outcomes was attenuated for low and high levels at t_0_ of each secondary outcome (insulin, glucose, and HbA1c). The ANCOVA model with post hoc testing showed that the interaction (group^x^secondary variable t_0_^2^) was significant for insulin with *p* = 0.007, glucose with *p* = 0.001 (Appendix A), and HbA1c with *p* = 0.012 (Table 2), but only at the level of 2 g CO/day + LI (insulin: EMM [95% CI]: −0.16 µU/L [−2.4–1.92], *p* = 0.002; glucose: EMM [95% CI]: −1.24 mg/dL [−3.76–1.28], *p* < 0.001; HbA1c: EMM [95% CI]: −0.02% [−0.08–0.04], *p* = 0.005) compared to the placebo. The effect of CO on secondary outcomes was evaluated at each level of the IR phenotype. The interaction of the group^x^IR phenotype was significant for the change of insulin levels with *p* = 0.025 but not for the change of glucose and HbA1c levels. Post hoc testing showed that the effects of CO on insulin levels occurred only in subjects with mild IR and only between the 4 g CO/day group and the placebo group (EMM [95% CI]: −2.49 µU/L [−5.14–0.17], *p* = 0.027).

The effect of CO on the parameters of inflammation and lipid metabolism are presented in Table 3. Before the intervention (t_0_), no significant difference existed. After 12 weeks of intervention, no different effect of CO on the parameters of inflammation and lipid metabolism could be observed.

## 3. Discussion

In the present study, CO supplementation, either alone or in combination with LI, showed a decrease in the HOMA index in obese subjects, with a significant effect observed when 2 g CO/day was combined with LI in the entire cohort. A significant reduction in the HOMA index without LI was observed at higher doses of 4 g CO/day in the subgroup of subjects with mild IR.

The combination of CO supplementation with LI resulted in a significant decrease in the HOMA index over all the IR phenotypes, probably due to the synergistic anti-inflammatory and anti-diabetic effects of CO + LI. However, the underlying biomolecular mechanism of such an interaction remains poorly understood [17,18].

However, pre-clinical studies in mice have shown that CO-derived wax esters decrease VAT and liver fat mass, and reduce macrophage infiltration into VAT, leading to an increase in adiponectin expression [15]. Subclinical inflammation originating from VAT, an imbalance in body fat distribution, or a combination of both, plays a crucial role in the pathophysiology of IR [19,20,21]. Previous studies investigating the effects of CO in combination with LI on body composition have shown positive results [22,23,24]. However, whether it also has potential effects on inflammation has not been investigated. Wasserfurth et al. [23] observed a significant reduction in body fat mass compared to a placebo; this was observed in response to 12 weeks of 2 g CO/day supplementation in combination with LI in 134 overweight adults (59.4 ± 5.6 years, BMI: 28.4 ± 5.8 kg/m^2^) with unlikely IR (initial HOMA index: 2.3 ± 1.8). Similarly, Dad’ová et al. [24] reported a reduction in VAT in overweight older women (70.9 ± 3.9 years, BMI: 27.2 ± 3.9 kg/m^2^) after a combination of 2.5 g CO/day +LI. In the present study, no changes in body composition markers were observed in any of the intervention groups (Appendix A: Parameters of body composition (t_0_) and after (t_12_) the intervention). The difference in response to body composition markers between the study by Wasserfurth et al., the study by Dad’ová et al., and the present study may be due to a different intensity of LI. While the Wasserfurth et al. study found an increase in self-reported baseline activity and the Dad’ová study found significant improvements in functional physical performance markers, this study found no changes in parameters of the 6 min walk test (Appendix A: Parameters of the 6-min walk test before (t_0_) and after (t_12_) the intervention) and self-reported regular physical activity (Appendix A: Parameters of regular physical activity before (t_0_) and after (t_12_) the intervention). It is also important to note the difference in BMI between the trials. In Wasserfurth and Dad’ová et al., the subjects had a significantly lower BMI (28.4 ± 5.8 kg/m^2^ and 27.2 ± 3.9 kg/m^2^, respectively) compared to the present study (34.6 ± 5.3 kg/m^2^). Among several other determinants, a higher initial BMI has been described as a barrier to behavioral change in LI [25]. In addition, the LI in Wasserfurth and Dad’ová et al. differed from this study in that it was led by a professional trainer, took place regularly in a gym and was progressively increased, which may explain the greater effect on weight loss.

In general, there is a debate about phenotyping the prediabetic population in order to improve the prevention of progression to diabetes [26,27]. Detailed, personalized information is essential for clinicians to better estimate the effectiveness of preventive strategies for different types of patients, leading to more targeted and effective intervention.

In the present study, significant effects of CO without LI were only seen in subjects with a HOMA index between 2.5 and 5. A previous study by Burhop et al. [16] found that 12 weeks of supplementation with 2 g CO/day in 43 obese subjects (BMI: 31.7 ± 5.2 kg/m^2^) with an initial HOMA index value of 4.1 ± 1.9, resulted in a significant reduction in the HOMA index and hepatic IR values. Subgroup analysis was not reported by Burhop et al., but the initial range of HOMA indices corresponded to the type of IR, for which we found a significant reduction in the HOMA index without LI in the 4 g CO/day group in the present study. This finding is significant for clinical considerations, as it highlights the variability in intervention effects and underscores the importance of tailoring CO interventions to individual patient profiles, particularly in those with mild IR. Dietary supplements containing CO may be beneficial in a preventive approach when the progression of IR is at an early stage. In severe IR, CO had no further benefit, as no therapeutic approach can be expected from dietary supplements. In such metabolic disorders, LI, including dietary supplements, may be underpowered.

In line with our findings, Abbort et al. [28] also found that n3 PUFA supplementation from fish oil (860 mg/day DHA + 120 mg/day EPA) had a significant lowering effect on the HOMA index only when the initial HOMA index was >2.5. A possible reason for the lack of effects on the HOMA index ranges < 2.5 may be that the underlying pathophysiology—such as unfavorable body fat distribution, particularly liver fat content, and levels of metabolic stress and pro-inflammatory mediators—is either absent or not sufficiently pronounced to necessitate modulation.

Conversely, severe IR (HOMA index > 5.0) is closely linked to significant metabolic dysregulation, including a higher prevalence of hypertension, hypercholesterolemia, and higher amounts of liver fat, ultimately increasing the risk T2DM [29]. A possible reason for the attenuated effects of CO supplementation in this study in subjects with a HOMA index >5.0 may be that, in cases of severe metabolic dysregulation, interventions with relatively low doses of n3 PUFAs may not be effective. Even at the highest dose of CO (4 g CO/day: 276 mg EPA + 256 mg DHA), the amounts of n3 PUFAs are relatively low, compared to other studies in subjects with severe IR. A study by Derosa et al. [30] observed a significant reduction in the HOMA index compared to the placebo in 281 overweight subjects with severe IR after 9 and 18 months of 3.0 g of n3 PUFA supplementation from fish oil in combination with a controlled energy diet. In contrast, the intervention in the CAGHO study was carried out without a controlled diet and for only 3 months, so we can only speculate whether higher doses of CO would also have significant effects in subjects with severe IR.

In addition to n3 PUFAs, several other bioactive compounds found in CO, such as astaxanthin, plant sterols, and fatty alcohols, likely possess anti-inflammatory and antioxidant properties, contributing to its effects on glucose homeostasis. In pre-clinical studies with obese mice, astaxanthin supplementation showed beneficial effects on insulin signaling and inflammatory processes [31]. However, clinical trials—generally using much higher doses of astaxanthin supplementation (ranging from 4 mg/day to 20 mg/day) compared to this study (in 4 g CO/day: 4 mg astaxanthin/day)—have failed to show a beneficial effect on glucose homeostasis [32,33]. Preliminary evidence from a recent clinical trial showed that the combination of n3 PUFA supplementation (1000 mg EPA/day + 400 mg DHA/day from fish oil) with plant sterols (1.7 g/day) had superior effects on glucose metabolism in overweight subjects (BMI: 26.2 ± 3.7 kg/m^2^) with mild IR (HOMA index: 2.9 ± 1.2) compared to n3 PUFA supplementation alone [34]. In the present study, even in the high-dose group (4 g CO/day), the doses administered were relatively low (ranking from 7 mg/day in 2 g CO/day to 14 mg/day in 4 g CO/day). Moreover, CO contains relatively high doses of fatty alcohols such as marine policosanols (4 g CO/day contains 1152 mg of policosanols). In a recent clinical trial involving 80 elderly dyslipidemic subjects, policosanol supplementation demonstrated beneficial effects on IR at significantly lower doses (10 mg/day) in combination with other components, such as astaxanthin, berberine, and coenzyme Q10 [13]. However, the study does not present any findings on the effects of pure policosanol.

### Limitations

This study has several limitations. Liver fat content, which plays a key role in characterizing IR phenotypes and predicting improved insulin secretion, was not assessed before or after the intervention. We did not observe any changes in body weight, body fat content or calculated visceral fat mass, which is highly correlated with liver fat content. Therefore, the influence of changes in liver fat content is likely to be marginal. Insulin sensitivity was not assessed by the oral glucose tolerance test (oGTT) or the hyperinsulinemic-euglycemic clamp technique, which reflect more detailed methods of assessing glucose intolerance and beta cell function, although alternative markers, such as fasting glucose, HbA1c, and the HOMA index were used instead. The use of the HOMA index is limited in subjects who do not have functioning beta cells. Additionally, the correlation between the HOMA index and the hyperinsulinemic-euglycemic clamp technique is reduced in normal weight populations. However, this study used obese population without T2DM or T1DM subjects. Additionally, the EPA + DHA blood status was not measured. The variability in the basal EPA + DHA blood status among test subjects, as well as the post-intervention response, might also impact the intervention’s effectiveness.

## 4. Materials and Methods

### 4.1. Study Design and Study Subjects

This exploratory study, referred to as the CAGHO (Calanus Oil for Glucose Homeostasis) study was conducted as a four-armed, double-blinded, and randomized clinical trial at the Institute of Food Science and Human Nutrition, Leibniz University Hannover, Germany and at the Division of Performance and Health (Sports Medicine) Institute for Sport and Sport Science, TU Dortmund University according to the guidelines of the Declaration of Helsinki and it was registered in the German Clinical Register (DRKS00030256). All participants gave informed consent prior to enrolment. Subjects were recruited through advertisements in the local press and announcements on social media platforms. Interested individuals were screened for eligibility using a digital screening questionnaire. After the verification of inclusion criteria [age between 30 and 75 years, abdominal obesity characterized by body mass index (BMI) ≥ 28 kg/m^2^, waist circumference (WC) ≥ 88 cm for women and ≥102 cm for men] and exclusion criteria (e.g., consumption of n3 FA supplements, severe or chronic diseases, pregnancy), subjects were invited to the study site for blood testing. After verifying additional blood-test-based inclusion criteria—including HOMA index ≥ 2.5, glucose from 100 mg/dL to 126 mg/dL, or HbA1c from 5.6% to 6.5%—a total of 266 subjects were enrolled in the study. Invited subjects were randomized—stratified by BMI, WC, sex, and age—into the following study groups (Figure 1):A total of 2 g CO/day spread over 4 capsules;A total of 4 g CO/day spread over 8 capsules;A total of 2 g CO/day in combination with LI (diet counselling + moderate exercise, approximately 2.5 h per week;Placebo, 4 capsules of paraffin oil per day.

The 2 g CO/day dosage has been used in previous studies and is the amount of CO usually to be consumed [16,23,24]. This study aimed to assess (a) the synergistic effects of 2 g CO/day with LI vs. 2 g CO/day alone and (b) potential dose-dependent effects, prompting the inclusion of a 4 g CO/day group.

### 4.2. Test Products

The composition of CO (Zooca^®^Lipids, Tromsø, Norway) is shown in Table 4. Subjects were advised to consume 4 (2 g CO/day, 2 g CO/day + LI and placebo) or 8 (4 g CO/day) capsules daily with water (at least 200 mL) and a meal to ensure adequate fat digestion and absorption [35]. To assess compliance, participants received a defined number of capsules, which were counted at the end of the intervention at the t_12_ examination. A subject was judged as compliant if at least 85% of the capsules were taken.

### 4.3. Anthropometric and Body Composition Measurements

Height was measured using a stadiometer (Seca GmbH & Co., KG, Hamburg, Germany). WC was measured between the lowest rib and the highest hip bone at the narrowest part of the midsection using a tape measure. Body weight was measured digitally (Seca GmbH & Co., KG, Hamburg, Germany) to the nearest 0.1 kg (lightly dressed, without shoes). The body composition markers relative to fat mass, phase angle, and visceral fat mass were analyzed using an 8-point bioelectrical impedance analyzer (BIA, mBCA525, Seca Company, Hamburg, Germany). Prior to measurement, subjects were instructed to urinate and remove all jewelry. Subjects were instructed to lie down on a stretcher and rest for about 5 min to ensure a balanced distribution of body fluids. All measurements were taken by trained nutritionists of the Institute.

### 4.4. Monitoring of Lifestyle Intervention and Physical Activity

Subjects in the 2 g CO/day + LI group were advised to engage in regular exercise in accordance with WHO recommendations [36]. Specifically, subjects were instructed to perform at least 150–300 min of moderate physical activity or at least 75–150 min of vigorous physical activity. To monitor LI adherence and weekly physical activity, subjects received a physical activity diary at t_0_ and t_12_. Those reporting <85% of the instructed activity per week, showing strong fluctuations between weeks, or achieving <85% overall were excluded from the analysis. Diet counseling included general recommendations from national professional societies for healthier eating.

In addition, the amount of regular physical activity during the intervention was recorded at t_0_ and t_12_ using the Freiburg Physical Activity Questionnaire described by Frey et al. [37]. On every examination day, subjects performed a 6 min walk test (6MWT) to assess their fitness level. The 6MWT was performed on a 30 m walking track, measuring the distance that subjects walked in 6 min [38]. Additionally, subjects were asked to rate their level of breathlessness on a scale from 1 to 10 immediately before and after the test. Pulse and oxygen saturation were also measured before and immediately after the test using a pulse oximeter. Subjects randomized in groups of 2 g CO/day, 4 g CO/day, and a placebo were instructed not to change their dietary habits (especially regarding the intake of n3 PUFA-rich foods) or physical activity during the intervention period to minimize dietary effects on variability in n3 PUFA status and glucose metabolism.

### 4.5. Blood Sampling and Biochemical Analysis

Blood samples were taken from the subjects after an overnight fast of at least 12 h, between 6:00 a.m. and 10:00 a.m., ideally at the same time on both examination days. Venipuncture was performed on an arm vein using EDTA tubes, serum tubes, and Gluco Exact tubes (Sarstedt AG & Co., KG, Nümbrecht, Germany). The samples were stored at 5 °C and transported on the same day to an accredited and certified laboratory (Laboratory Group Dr. Kramer and Colleagues). Glucose levels were assessed using a photometric method (Beckman Coulter GmbH, Krefeld, Germany). Insulin levels were measured by using an electrochemiluminescence immunoassay (ECLIA) with the cobas 801e system (Roche Diagnostics GmbH, Mannheim, Germany), while HbA1c was determined using high-pressure liquid chromatography (HPLC) (Tosoh Bioscience, Griesheim, Germany). Triacylglycerols (TAG), high-density lipoprotein cholesterol (HDL-C), and low-density lipoprotein cholesterol (LDL-C) were analyzed from serum tubes using a photometric method (Beckman Coulter GmbH, Germany).

The browser-based American Metabolic Syndrome (MetS) Severity Calculator (https://metscalc.org/metscalc/, accessed on 12 August 2024) was used to calculate MetS severity scores for each subject. The calculated MetS severity score was first described by Gurka and De Boer et al. [39] and takes into account the following cardiovascular disease risk parameters: systolic blood pressure, TG, HDL-C, fasting glucose, as well as information on sex, age, race/ethnicity, and weight. As a result, a single value based on BMI and a single value based on WC were calculated for each person.

The HOMA index was selected as the primary outcome and was calculated according to the method of Matthews et al. [5] as follows:(1)$$\mathrm{HOMA}\;\mathrm{index}=\frac{\mathrm{fasting}\;\mathrm{insulin}\;\left(\frac{\mu U}{\mathrm{mL}}\right)\times \mathrm{fasting}\;\mathrm{glucose}\;\left(\frac{\mathrm{mg}}{\mathrm{dL}}\right)}{405}$$

The HOMA index was used as the primary outcome instead of fasting glucose and fasting insulin because the HOMA index measures IR by assuming feedback between the liver and beta cells. Glucose concentrations are regulated by insulin-dependent glucose production in the liver, while insulin levels depend on the response of pancreatic beta cells to glucose concentrations. Therefore, a reduced response to glucose-stimulated insulin secretion reflects a deficiency in beta cell function. IR can be observed by the reduced suppressive effect of insulin on glucose production in the liver.

To assess the effects of CO in relation to different degrees of IR, subjects were categorized into IR phenotypes based on their HOMA index prior to the intervention (t_0_). The classification included three phenotypes: I (unlikely IR), II (mild IR), and III (severe IR). Phenotype I, indicating unlikely IR, was defined as a HOMA index < 2.5, based on the thresholds suggested by Keskin et al. [40], Singh et al. [8], and Lee et al. [9]. Phenotype II, representing mild IR, was defined by a HOMA index range from 2.5 to 5.0, consistent with the findings from Wang et al. [41]. For phenotype III, indicating severe IR, a HOMA index > 5.0 was used. This threshold is suggested for severe IR by Lee et al. [42].

### 4.6. Sample Size Calculation and Statistical Analysis

The sample size calculation was based on a first-in-human study, where the “CO group” was similar to the 2 g CO/day group in this study [16]. Assuming no change and similar SD in the placebo group, an effect size of 0.70 was expected at 12 weeks. To detect this with 80% power and 5% significance, 35 patients per group were needed. However, due to high SDs in the HOMA index, the effect size was reduced to 0.55, requiring 56 patients per group. With a 15% drop-out rate, 66 patients per group (total 264) were needed.

Statistical analysis was performed using SPSS software (IBM SPSS Statistics28.0; Chicago, IL, USA). Per-protocol analysis was used to assess the effects of the intervention only in participants who adhered to the study protocol. This approach minimized bias from non-compliance and dropouts, providing a clearer understanding of the potential of the intervention under ideal conditions. Continuous variables are presented as mean and standard deviation (SD), while qualitative variables are presented either as absolute or relative frequencies. The Shapiro–Wilk test and visual inspection of quantile–quantile plots were used to test for normal distribution of continuous variables. The chi-squared test was used to determine the distribution of nominal variables between the groups. Normally distributed data were tested using univariate one-way analysis of variance (ANOVA) with Least Significant Difference (LSD) correction for the post hoc test to assess differences in baseline characteristics between groups and glucose metabolism parameters before the intervention (t_0_). Due to the exploratory nature of this study, a simple slope analysis (Figure 2) including main group effects, linear, and squared interactions with the HOMA index at t_0_ was used to identify areas of IR that were most responsive to CO supplementation. To determine the effects of CO supplementation, the analysis of covariance (ANCOVA) was performed with a change in the HOMA index, insulin, glucose, or HbA1c (Δt_12_ − t_0_) as dependent variables. The study group was used as a fixed factor. Centered measurement of the HOMA index, insulin, glucose and HbA1c before the intervention (t_0_), squared measurement of HOMA index, insulin, glucose, HbA1c at t_0_, BMI, age, and sex were used as covariates. Squared interaction was entered because, based on simple slope analysis, a nonlinear moderator effect of initial HOMA index levels on the effect of intervention was suspected. Age and BMI were used as covariates because age and BMI were significantly associated with measurements of the HOMA index, insulin, glucose, and HbA1c at t_0_. If significant, a post hoc test was performed to determine the difference between the study groups. As an alternative approach, the effects of CO supplementation across different IR phenotypes were analyzed using ANCOVA with change in the HOMA index (Δt_12_ − t_0_) as the dependent variable and group as the fixed factor. IR phenotype classification as a categorical variable, BMI, age, and sex were used as covariates. A post hoc test (LSD) was used to test for simple main effects at each level of the IR phenotype. The statistically significant level of *p* < 0.05 was used for all analyses.

## 5. Conclusions

Our data suggest that individuals with mild IR may benefit from CO supplementation, either at a high dose or at a lower dose in combination with LI, potentially leading to improved insulin sensitivity. The clinical relevance of these exploratory findings needs to be clarified through long-term follow-up studies. Additionally, further research is needed to explore the impact of high doses of CO when combined with LI.

## Figures and Tables

**Figure 1 marinedrugs-23-00139-f001:**
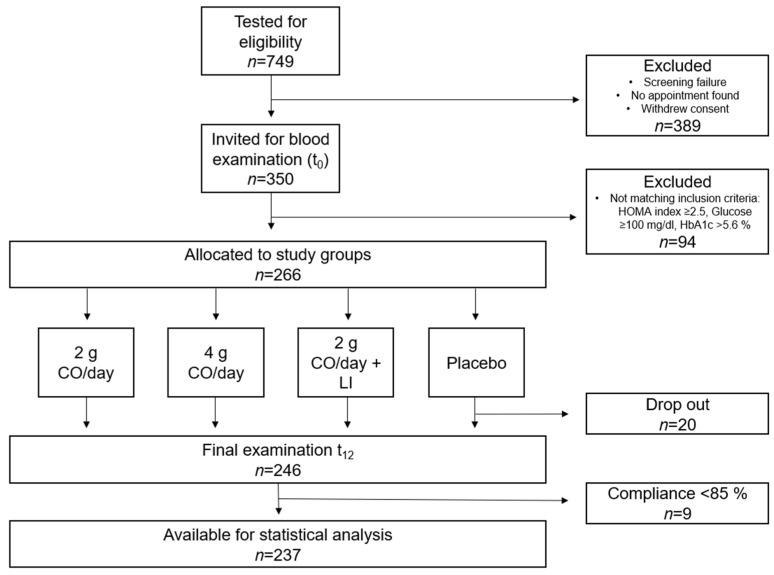
Flow diagram of the study population.

**Figure 2 marinedrugs-23-00139-f002:**
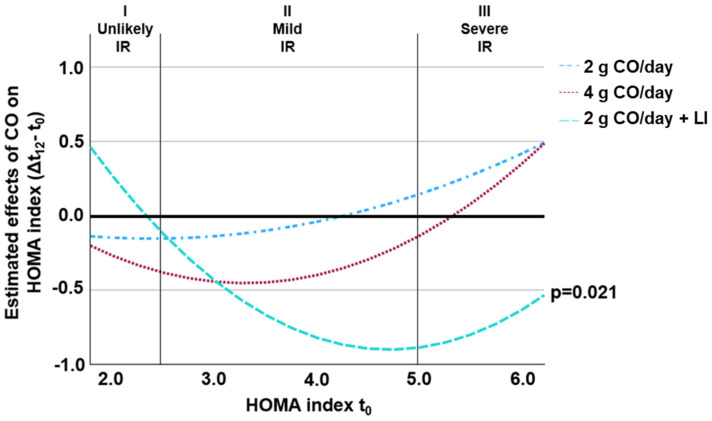
Estimated effects of CO on the change in the HOMA index (Δt_12_ − t_0_) compared to placebo across initial (t_0_) HOMA index.

**Figure 3 marinedrugs-23-00139-f003:**
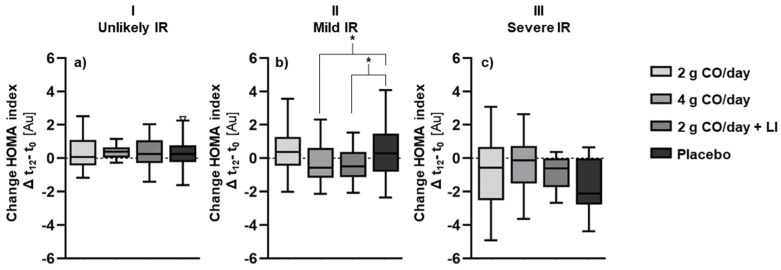
Changes in HOMA index between study groups: (**a**) in subjects with IR phenotype: I (unlikely), (**b**) in subjects with IR phenotype: II (mild), (**c**) in subjects with IR phenotype III: (severe). * Significantly different between the groups at *p* < 0.05.

**Table 1 marinedrugs-23-00139-t001:** Study cohort characteristics at baseline (study cohort as per protocol).

	2 g CO/Day *n* = 61	4 g CO/Day *n* = 61	2 g CO/Day + LI *n* = 57	Placebo *n* = 58	*p*-Value
**Variables**	n (%)	n (%)	n (%)	n (%)	
**Gender**					
Female	41 (68)	42 (69)	39 (68)	38 (66)	0.981 ^a^
Male	20 (32)	18 (31)	18 (32)	20 (34)	
**Anthropometric**	Mean ± SD	Mean ± SD	Mean ± SD	Mean ± SD	
Age [year]	54.2 ± 9.6	57.3 ± 11.3	56.0 ± 9.0	54.4 ± 9.4	0.265 ^b^
BMI [kg/m^2^]	34.2 ± 5.4	34.4 ± 4.1	34.2 ± 6.1	34.1 ± 4.3	0.990 ^b^
WC [cm]	109 ± 12.6	111 ± 10.6	110 ± 13.4	111 ± 11.2	0.815 ^b^
**Body composition**	Mean ± SD	Mean ± SD	Mean ± SD	Mean ± SD	
Phase angle [°]	6.0 ± 0.8	5.8 ± 0.8	6.0 ± 0.7	5.8 ± 0.7	0.282 ^b^
Body fat [%]	42.4 ± 7.3	42.8 ± 7.5	41.6 ± 8.0	43.0 ± 7.6	0.782 ^b^
Visceral fat mass [L]	4.2 ± 2.5	4.2 ± 212	4.2 ± 2.7	4.1 ± 2.2	0.994 ^b^
**Metabolic Syndrome Severity (MetS) Score**					
MetS score (based on BMI)	0.6 ± 0.6	0.6 ± 0.5	0.6 ± 0.6	0.5 ± 0.5	0.931 ^b^
MetS score (based on WC)	0.7 ± 0.8	0.6 ± 0.5	0.6 ± 0.6	0.6 ± 0.5	0.845 ^b^

Abbreviations: BMI, body mass index; CO, calanus oil; LI, lifestyle intervention; SD, standard deviation; WC, waist circumference. ^a^ Chi-squared test. ^b^ One-way ANOVA.

**Table 2 marinedrugs-23-00139-t002:** Parameters of glucose metabolism before (t_0_) and after (t_12_) the intervention.

	2 g CO/Day	4 g CO/Day	2 g CO/Day + LI	Placebo	*p*-Value Effect Size		
	*n* = 61	*n* = 61	*n* = 57	*n* = 58			
Variables	Mean ± SD				At t_0_ ^a^	Interaction ^b^	η_p_^2^
**HOMA index** [AU]							
t_0_	3.9 ± 2.0	4.0 ± 2.1	4.0 ± 2.0	3.8 ± 1.7	0.900		
t_12_	4.0 ± 2.2	4.0 ± 2.7	3.8 ± 2.2	3.9 ± 2.3		**0.011 ***	0.043
**HbA1c** [%]							
t_0_	5.6 ± 0.3	5.6 ± 0.3	5.6 ± 0.3	5.7 ± 0.3	0.619		
t_12_	5.6 ± 0.3	5.6 ± 0.3	5.6 ± 0.3	5.7 ± 0.3		**0.014 ***	0.004

Abbreviations: AU, arbitrary unit; CO, calanus oil; HOMA index, homeostatic model assessment of insulin resistance; LI, lifestyle intervention; SD, standard deviation; η_p_^2^ partial eta square. Significant *p*-values (*p* < 0.05) are shown in bold. ^a^ Difference before the intervention at t_0_: One-way ANOVA. ^b^ ANCOVA with change in HOMA index; HbA1c (Δt_12_ − t_0_) as dependent variable, study group as a fixed factor and centered variables at t_0_, squared variables at t_0_, BMI, age, and sex as covariates. Significant difference in post hoc tests: * Group 2 g CO/day + LI vs. placebo group.

**Table 3 marinedrugs-23-00139-t003:** Parameters of inflammation and lipid metabolism before (t_0_) and after (t_12_) the intervention.

	2 g CO/Day	4 g CO/Day	2 g CO/Day + LI	Placebo	*p*-Value Effect Size		
	*n* = 61	*n* = 61	*n* = 57	*n* = 58			
Variables	Mean ± SD				At t_0_ ^a^	Interaction ^b^	η_p_^2^
**Inflammation**							
**CRP** [mg/dL]							
t_0_	3.1 ± 3.6	3.4 ± 7.4	3.7 ± 4.6	4.4 ± 6.9	0.626		
t_12_	2.8 ± 2.9	2.9 ± 3.0	3.5 ± 3.3	3.6 ± 3.5		0.318	0.016
**Lipid metabolism**							
**TG** [mg/dL]							
t_0_	151 ± 70.7	146 ± 68.2	140 ± 57.9	128 ± 64.9	0.176		
t_12_	139 ± 65.0	141 ± 61.8	133 ± 57.2	136 ± 61.2		0.092	0.027
**TC** [mg/dL]							
t_0_	232 ± 43.0	227 ± 37.2	238 ± 48.7	226 ± 43.5	0.470		
t_12_	232 ± 43.4	229 ± 43.2	231 ± 42.7	227 ± 46.1		0.147	0.023
**HDL-C** [mg/dL]							
t_0_	57.5 ± 11.9	58.7 ± 13.9	61.5 ± 11.6	59.8 ± 12.3	0.154		
t_12_	56.2 ± 11.6	57.7 ± 14.6	58.8 ± 11.5	57.8 ± 12.0		0.425	0.012
**LDL-C** [mg/dL]							
t_0_	148 ± 30.2	142 ± 27.1	151 ± 36.1	142 ± 31.6	0.426		
t_12_	148 ± 32.7	140 ± 30.9	145 ± 31.0	142 ± 34.1		0.256	0.017

Abbreviations: CO, calanus oil; TG, triglyceride; TC, total cholesterol; HDL-C, high density lipoprotein cholesterol; LDL-C, low density lipoprotein cholesterol; SD, standard deviation; η_p_^2^ partial eta square. ^a^ Difference before the intervention at t_0_: one-way ANOVA. ^b^ ANCOVA with change in CRP; TG; TC; LDL-C; HDL-C (Δt_12_ − t_0_) as dependent variable, study group as a fixed factor and BMI, age and sex as covariates. Interaction reflects the group ^x^ time effect.

**Table 4 marinedrugs-23-00139-t004:** Composition of calanus oil.

Components	g/ 100 g CO	mg/ 2 g CO	mg/ 4 g CO
MUFA	9.7	194	388
PUFA	26.2	524	1048
n3 PUFAs	25.0	500	1000
ALA	1.4	28	56
SDA	8.4	168	336
EPA	6.9	138	276
DHA	6.4	128	256
n6 PUFAs	1.1	22	44
LA	0.7	14	28
ARA	0.2	4	8
Fatty alcohols	28.8	576	1152
Sterols	0.35	7	14
Astaxanthin	0.1	2	4

Abbreviations: ALA, alpha-linolenic acid; ARA, arachidonic acid; CO, calanus oil; DHA, docosahexaenoic acid; EPA, eicosapentaenoic acid; LA, linoleic acid; MUFA, monounsaturated fatty acids; PUFA, polyunsaturated fatty acids; SDA, stearidonic acid.

## Data Availability

Data are available upon request from the corresponding author.

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
