# Peer review of "Calanus Oil and Lifestyle Interventions Improve Glucose Homeostasis in Obese Subjects with Insulin Resistance"

_marinedrugs, 2025, doi:10.3390/md23040139_

Round 1
Reviewer 1 Report
Comments and Suggestions for Authors
The manuscript detailed the experiments and the results appropriately but had some issues, which are mentioned below:
The abstract is fine, but more information about the composition of CO and future directions of the study needs to be added.
The introduction section lacks recent information in the field, especially the importance of dietary restrictions and other possible obesity and DM management methods.
Highlight the importance of CO in the introduction.
Results are expressed in a scientifically accepted way.
The discussion part is very weak; discuss the results with previous reports not only on CO.
“A post hoc test showed significant improvement with 2 g CO/d + LI (p=0.021). Secondary analysis revealed that 4 g CO/d had significant effects in subjects with mild IR (HOMA index 2.5-5.0) (p=0.043)”. According to this statement, LI has a major role. CO supplementation doesn’t seem to have much influence. So, explain the rationale for grouping. Another group of 4 g CO/d in combination with LI is missing.
It is advised to remove the “4 g CO/d spread over 8 capsules from the study” or further studies are needed with “4 g CO/d in combination with LI” group.
Conclusion needs improvisation.
Some of the minor issues:
Line 15-18: Revise the sentence for clarity.
Line 23-25: What may be the reason for this?
Line 56: Italicize the scientific names.
Line 57, 58: What is FA, FAs?
Author Response
1. The abstract is fine, but more information about the composition of CO and future directions of the study needs to be added
More information on the CO composition has been added to the abstract (lines 13-15): “We investigated the effect of 12-weeks of Calanus oil (CO) supplementation at varying doses, combined with LI, on glucose homeostasis in obese subjects. CO is a novel marine oil containing long-chain omega-3 fatty acids in the form of wax esters as well as plant sterols and astaxanthin”.
2. The introduction section lacks recent information in the field, especially the importance of dietary restrictions and other possible obesity and DM management methods.
We have added some information on this in the introduction. We now state in lines 50-52: “Lifestyle interventions (LI), including dietary restrictions and increased physical activity, alongside pharmacotherapy, can result in clinically significant weight loss and enhanced insulin sensitivity”.
3. Highlight the importance of CO in the introduction.
Thanks for your comment, this has now been added to the introduction in lines 59-62: “Calanus oil (CO), derived from the marine crustacean Calanus finmarchicus, differs from conventional n-3 PUFA-rich oils in the binding form of its fatty acids (FAs). In CO, more than 80% of FAs are bound as wax esters, whereas in fish oil, they are primarily bound to triacylglycerols, and in krill oil, to phospholipids [12]. Furthermore, CO contains antioxidants like astaxanthin, as well as plant sterols and fatty alcohols, which may contribute to its effects”.
4. The discussion part is very weak; discuss the results with previous reports not only on CO.
Comparing the study results with the literature is challenging because a) CO's chemical composition differs significantly from other marine n3-rich oils, and b) there is limited knowledge about the effects of n3 on IR in the context of obesity. However, we have discussed several studies addressing the effect of n3 PUFA supplementation from fish oil alone or in combination with other possible antidiabetic agents:
- Lines 226-232: A study by Abbort et al. [1] is discussed, where 860 mg/d DHA + 120 mg/d EPA from fish oil were supplemented and a significant lowering effect on HOMA index was observed only when initial HOMA index was >2.5.
- Lines 240-246: A study by Derosa et al. [2] is discussed, where a significant reduction in HOMA index compared to placebo was observed in 281 overweight subjects with severe IR after 9 and 18 months of 3000 mg of n3 PUFA supplementation from fish oil in combination with a controlled energy diet.
- Lines 251-253 discuss a systematic review that looked at astaxanthin supplementation alone [3].
- Lines 255-258 discuss a clinical trial of 1000 mg EPA/d + 400 mg DHA/d from fish oil in combination with plant sterols on parameters of glucose metabolism [4].
- Lines 261-265: A clinical trial is mentioned that observed beneficial effects on IR of policosanol supplementation [5]
5. A post hoc test showed significant improvement with 2 g CO/d + LI (p=0.021). Secondary analysis revealed that 4 g CO/d had significant effects in subjects with mild IR (HOMA index 2.5-5.0) (p=0.043)”. According to this statement, LI has a major role. CO supplementation doesn’t seem to have much influence. So, explain the rationale for grouping. Another group of 4 g CO/d in combination with LI is missing
Thanks for your comment. This was not clearly stated.
Rationale for grouping:
To explain the study design in detail, it is important to provide some background. The effects of CO on glucose metabolism remain largely unexplored and inconsistent, with most studies in this field used 2 g CO/d alone or in combination with LI [6–10]. This dose is in line with the amount of CO usually to be consumed. However, 2 g CO/d only provides 138 mg EPA and 128 mg DHA, which may not have a clinically significant effect on the omega-3 index [11]. In addition, the International Society for the Study of Fatty Acids and Lipids (ISSFAL) recommends at least 500 mg of EPA and DHA per day [12], which is only covered by 4 g CO/d. For this reason, we are investigating possible dose-dependent effects, which led to the inclusion of a 4 g CO/d group. In our previous study, we used 2 g/d + LI in overweight but metabolically healthy subjects and found no improvement in IR. In this study, we investigate synergistic effects of 2 g CO/d in combination with LI compared to 2 g CO alone in obese subjects with different phenotypes of IR.
Although it would have been valuable to study an additional 4 g CO/d + LI group, this would have required a disproportionate increase in sample size, making further study arms infeasible.
A corresponding explanation was added in the methods section (lines 309-312): “The 2 g CO/d dosage has been used in previous studies and the amount of CO usually to be consumed [7,8,10]. This study aimed to assess a) the synergistic effects of 2 g CO/d with LI vs 2 g CO/d alone, and b) potential dose-dependent effects, prompting the inclusion of a 4 g CO/d group.”
According to this statement, LI has a major role. CO supplementation doesn’t seem to have much influence:
Our results indicate that both the 2 g CO/d + LI group and the 4 g CO/d group lead to a significant reduction in the HOMA index but only on the level of mild IR. For a better understanding of the results we now state in lines 218-225: “This finding is significant for clinical considerations, as it highlights the variability in intervention effects and underscores the importance of tailoring CO interventions to individual patient profiles, particularly in those with mild IR. Dietary supplements containing CO may be beneficial in a preventive approach when the progression of IR is at an early stage. In severe IR, CO had no further benefit, as no therapeutic approach can be expected from dietary supplements. In such metabolic disorders, LI, including dietary supplements, may be underpowered”.
6. It is advised to remove the “4 g CO/d spread over 8 capsules from the study” or further studies are needed with “4 g CO/d in combination with LI” group.
Our study follows the Declaration of Helsinki guidelines and is registered in a Clinical Register (DRKS00030256), prohibiting the removal of a study arm from the evaluation. However, we fully agree with the reviewer that further studies on 4 g CO/d with LI are necessary and have reflected this in our conclusion.
7. Conclusion needs improvisation
We have followed the reviewer’s suggestion and improved the conclusion. We now state in lines 432-436: “Our data suggest that individuals with mild IR may benefit from CO supplementation, either at a high dose or at a lower dose in combination with LI, potentially leading to improved insulin sensitivity. The clinical relevance of these exploratory findings needs to be clarified through long-term follow-up studies”.
8. Minor issues:
Line 15-15: Revise the sentence for clarity.
We revised lines 13-18 as you suggest. We now state: “We investigated the effect of Calanus oil (CO) supplementation, combined with LI, on glucose homeostasis in obese subjects”.
Line 23-25: What may be the reason for this?
This is now discussed in lines 176-178: “The combination of CO supplementation with LI resulted in a significant decrease in the HOMA index over all IR phenotypes, probably due to the synergistic anti-inflammatory and anti-diabetic effects of CO + LI. However, the underlying biomolecular mechanism of such interaction, remain poorly understood [13]”.
And in lines 218-225: “This finding is significant for clinical considerations, as it highlights the variability in treatment effects and underscores the importance of tailoring CO interventions to individual patient profiles, particularly in those with mild IR. Dietary supplements containing CO may be beneficial in a preventive approach when the progression of IR is at an early stage. In severe IR, CO had no further benefit, as no therapeutic approach can be expected from dietary supplements. In such metabolic disorders, LI, including dietary supplements, may be underpowered”.
Line 56: Italicize the scientific names
We italicize the scientific name as you suggest. Now in line 57.
Line 57, 58: What is FA, FAs
This was incorrectly stated. We changed to the plural which is correct: FAs.
References
[1] K.A. Abbott, T.L. Burrows, S. Acharya, R.N. Thota, M.L. Garg, DHA-enriched fish oil reduces insulin resistance in overweight and obese adults, Prostaglandins, leukotrienes, and essential fatty acids 159 (2020) 102154.
[2] G. Derosa, A.F.G. Cicero, A. D'Angelo, C. Borghi, P. Maffioli, Effects of n-3 pufas on fasting plasma glucose and insulin resistance in patients with impaired fasting glucose or impaired glucose tolerance, BioFactors (Oxford, England) 42 (2016) 316–322.
[3] W. Xia, N. Tang, H. Kord-Varkaneh, T.Y. Low, S.C. Tan, X. Wu et al., The effects of astaxanthin supplementation on obesity, blood pressure, CRP, glycemic biomarkers, and lipid profile: A meta-analysis of randomized controlled trials, Pharmacological research 161 (2020) 105113.
[4] J.-F. Wang, H.-M. Zhang, Y.-Y. Li, S. Xia, Y. Wei, L. Yang et al., A combination of omega-3 and plant sterols regulate glucose and lipid metabolism in individuals with impaired glucose regulation: a randomized and controlled clinical trial, Lipids in health and disease 18 (2019) 106.
[5] G. Marazzi, L. Cacciotti, F. Pelliccia, L. Iaia, M. Volterrani, G. Caminiti et al., Long-term effects of nutraceuticals (berberine, red yeast rice, policosanol) in elderly hypercholesterolemic patients, Advances in therapy 28 (2011) 1105–1113.
[6] T. Čížková, M. Štěpán, K. Daďová, B. Ondrůjová, L. Sontáková, E. Krauzová et al., Exercise Training Reduces Inflammation of Adipose Tissue in the Elderly: Cross-Sectional and Randomized Interventional Trial, The Journal of clinical endocrinology and metabolism 105 (2020).
[7] K. Daďová, M. Petr, M. Šteffl, L. Sontáková, M. Chlumský, M. Matouš et al., Effect of Calanus Oil Supplementation and 16 Week Exercise Program on Selected Fitness Parameters in Older Women, Nutrients 12 (2020).
[8] K.S. Tande, T.D. Vo, B.S. Lynch, Clinical safety evaluation of marine oil derived from Calanus finmarchicus, Regulatory toxicology and pharmacology RTP 80 (2016) 25–31.
[9] M. Burhop, J.P. Schuchardt, J. Nebl, M. Müller, R. Lichtinghagen, A. Hahn, Marine Oil from C. finmarchicus Enhances Glucose Homeostasis and Liver Insulin Resistance in Obese Prediabetic Individuals, Nutrients 14 (2022).
[10] P. Wasserfurth, J. Nebl, J.P. Schuchardt, M. Müller, T.K. Boßlau, K. Krüger et al., Effects of Exercise Combined with a Healthy Diet or Calanus finmarchicus Oil Supplementation on Body Composition and Metabolic Markers-A Pilot Study, Nutrients 12 (2020).
[11] M. Dempsey, M.S. Rockwell, L.M. Wentz, The influence of dietary and supplemental omega-3 fatty acids on the omega-3 index: A scoping review, Frontiers in nutrition 10 (2023) 1072653.
[12] International Society for the Study of Fatty Acids and Lipids (ISSFAL). PUFA Recommendations. Available online: http://www.issfal.org/statements/pufa-recommendations
[13] L.K. Oharomari, M.J. Ikemoto, D.J. Hwang, H. Koizumi, H. Soya, Benefits of Exercise and Astaxanthin Supplementation: Are There Additive or Synergistic Effects?, Antioxidants 10 (2021).
[14] L.-C. Tao, J.-N. Xu, T.-T. Wang, F. Hua, J.-J. Li, Triglyceride-glucose index as a marker in cardiovascular diseases: landscape and limitations, Cardiovascular diabetology 21 (2022) 68.
[15] E.S. Kang, Y.S. Yun, S.W. Park, H.J. Kim, C.W. Ahn, Y.D. Song et al., Limitation of the validity of the homeostasis model assessment as an index of insulin resistance in Korea, Metabolism: clinical and experimental 54 (2005) 206–211.
[16] K.-E. Eilertsen, H.K. Mæhre, I.J. Jensen, H. Devold, J.O. Olsen, R.K. Lie et al., A wax ester and astaxanthin-rich extract from the marine copepod Calanus finmarchicus attenuates atherogenesis in female apolipoprotein E-deficient mice, The Journal of nutrition 142 (2012) 508–512.
[17] E. Burgess, P. Hassmén, K.L. Pumpa, Determinants of adherence to lifestyle intervention in adults with obesity: a systematic review, Clinical obesity 7 (2017) 123–135.
[18] WHO Guidelines on Physical Activity and Sedentary Behaviour, 1st ed., World Health Organization, Geneva, 2020.
[19] M.D. DeBoer, S.L. Filipp, M.J. Gurka, Use of a Metabolic Syndrome Severity Z Score to Track Risk During Treatment of Prediabetes: An Analysis of the Diabetes Prevention Program, Diabetes care 41 (2018) 2421–2430.
[20] D.L. Tahapary, L.B. Pratisthita, N.A. Fitri, C. Marcella, S. Wafa, F. Kurniawan et al., Challenges in the diagnosis of insulin resistance: Focusing on the role of HOMA-IR and Tryglyceride/glucose index, Diabetes & metabolic syndrome 16 (2022) 102581.
Reviewer 2 Report
Comments and Suggestions for Authors
1. Shorten the title while maintaining key elements, for example, Calanus Oil and Lifestyle Intervention Improve Glucose Homeostasis in Insulin-Resistant Obese Individuals.
2. In the Abstract section, including quantitative results such as absolute HOMA-IR reduction and percentage improvements.
3. Overly detailed sections on VAT and lipotoxicity mechanisms could be condensed in the Introduction section.
4. No explicit hypothesis statement at the end of the Introduction section.
5. Explain why 2 g and 4 g doses were chosen.
6. Detail how compliance with LI was monitored.
7. Address HOMA-IR limitations and potential alternative assessments.
8. Is CO safe? What is the recommended dosage for humans? Is it not tested on animals? What is the FDA's safe dosage?
9. Include effect sizes for glucose, insulin, and HOMA-IR changes.
10. Provide additional metabolic markers (such as inflammatory markers, lipidomic changes) for mechanistic insights.
11. No stratification by sex (only in Table S4) or age groups.
12. Discuss why no weight loss was observed despite LI.
13. Provide practical implications for dietary supplementation in obesity management.
14. Suggest longer follow-up studies or alternative insulin sensitivity assessments.
15. Discussion should include more clinical implications.
Comments on the Quality of English Language
The English could be improved to more clearly express the research.
Author Response
1. Shorten the title while maintaining key elements, for example, Calanus Oil and Lifestyle Intervention Improve Glucose Homeostasis in Insulin-Resistant Obese Individuals.
We follow your suggestion and changed the title accordingly: “Calanus Oil and Lifestyle Intervention Improve Glucose Homeostasis in Obese Subjects with Insulin Resistance”
2. In the Abstract section, including quantitative results such as absolute HOMA-IR reduction and percentage improvements.
The reviewer is right. We carefully revised the abstract and include quantitative results. We now state in lines 19-23: “Post hoc test showed significant improvement with 2 g CO/d + LI (estimated marginal means [EMM] 95% confidence interval [CI]: -0.19 [-0.80 - 0.41], p=0.021). Secondary analysis revealed that 4 g CO/d had significant effects in subjects with mild IR (HOMA index 2.5-5.0) (EMM [95 % CI]: -0.76 [-1.53 – 0.03], p=0.043). CO supplementation improved glucose homeostasis, with effects varying by dose, combination with LI and IR phenotype”.
3. Overly detailed sections on VAT and lipotoxicity mechanisms could be condensed in the Introduction section
This was inappropriately detailed and we have reduced it as you suggested and now state in lines 33-36: “The hypertrophy of visceral and ectopic adipocytes induces chronic inflammation that impairs peripheral insulin action [3]. Additionally, VAT serves as a marker for ectopic lipid accumulation, leading to the formation of lipotoxic lipids like diacylglycerols and ceramides, which inhibit insulin signaling”.
However, it is important to avoid any misinterpretation that obesity leads to a pathophysiological impairment of insulin sensitivity solely through the inflammatory pathway.
4. No explicit hypothesis statement at the end of the Introduction section.
The reviewer is right. In lines 69-70, we added our hypothesis: “We hypothesized that CO supplementation, both alone or combined with LI, could improve IR in subjects with abdominal obesity. Therefore, we aimed to examine the effects of 12-weeks of CO supplementation at varying doses, with or without LI, on parameters of glucose metabolism, focusing specifically on HOMA index in obese subjects with IR”.
5. Explain why 2 g and 4 g doses were chosen
Rationale for grouping:
To explain the study design in detail, it is important to provide some background. The effects of CO on glucose metabolism remain largely unexplored and inconsistent, with most studies in this field used 2 g CO/d alone or in combination with LI [6–10]. This dose is in line with the amount of CO usually to be consumed. However, 2 g CO/d only provides 138 mg EPA and 128 mg DHA, which may not have a clinically significant effect on the omega-3 index [11]. In addition, the International Society for the Study of Fatty Acids and Lipids (ISSFAL) recommends at least 500 mg of EPA and DHA per day [12], which is only covered by 4 g CO/d. For this reason, we are investigating possible dose-dependent effects, which led to the inclusion of a 4 g CO/d group. In our previous study, we used 2 g/d + LI in overweight but metabolically healthy subjects and found no improvement in IR. In this study, we investigate synergistic effects of 2 g CO/d in combination with LI compared to 2 g CO alone in obese subjects with different phenotypes of IR.
Although it would have been valuable to study an additional 4 g CO/d + LI group, this would have required a disproportionate increase in sample size, making further study arms infeasible.
A corresponding explanation was added in the methods section (lines 309-312): “The 2 g CO/d dosage has been used in previous studies and the amount of CO usually to be consumed [7,8,10]. This study aimed to assess a) the synergistic effects of 2 g CO/d with LI vs 2 g CO/d alone, and b) potential dose-dependent effects, prompting the inclusion of a 4 g CO/d group”.
6. Detail how compliance with LI was monitored.
Thanks for having an eye on this. The information was missing. We have carefully revised section 4.4 in the Methods and included more information about monitoring LI and physical activity. This is now included in lines: 339-342:
“To monitor LI adherence and weekly physical activity, subjects received a physical activity diary at the initial and final examination. Those reporting <85% of the instructed activity per week, showing strong fluctuations between weeks, or achieving <85% overall were excluded from the analysis”.
7. Address HOMA-IR limitations and potential alternative assessments.
We kindly thanks for your comment. We discuss the limitations of the HOMA index and alternative assessment methods of glucose impairment in lines 268-279: “This study has several limitations. Liver fat content, which plays a key role in characterizing IR phenotypes and predicting improved insulin secretion, was not assessed. As we did not observe any changes in body weight, body fat content or calculated visceral fat mass, which is highly correlated with liver fat content. Therefore, the influence of changes in liver fat content is likely to be marginal. Insulin sensitivity was not assessed by the oral glucose tolerance test (oGTT) or the hyperinsulinemic-euglycemic clamp technique, which reflect more detailed methods of assessing glucose intolerance and beta-cell function, although alternative markers such as fasting glucose, HbA1c and the HOMA index were used instead. The use of the HOMA index is limited in subjects who do not have functioning beta cells [14]. Additionally, the correlation between the HOMA index the hyperinsulinemic-euglycemic clamp technique is reduced in normal weight populations [15]. However, this study used obese population without T2DM or T1DM subjects”.
8. Is CO safe? What is the recommended dosage for humans? Is it not tested on animals? What is the FDA's safe dosage?
The safety of CO has been proven in both preclinical [16] and clinical studies[8]. Moreover, CO has received Generally Recognized as Safe (GRAS) status from the U.S. Food and Drug Administration (FDA).
9. Include effect sizes for glucose, insulin, and HOMA-IR changes.
The reviewer is right. We have included the effect sizes (partial eta squared) for glucose, insulin and HOMA index in Table 2
10. Provide additional metabolic markers (such as inflammatory markers, lipidomic changes) for mechanistic insights
We thank for your suggestion and include CRP as a marker of inflammation and parameters of lipid metabolism in a new table, now Table 3.
11. No stratification by sex (only in Table S4) or age groups.
There is no difference in sex distribution or age between the groups (Table 1). We also did not perform sex or age group stratified analyses. Rather, we included sex as a covariate in all statistical models evaluating the effect of CO on primary and secondary parameters. No different results were observed.
Age as a metric variable was also included as a covariate in all statistical models. However, age was not used as a categorical variable using age groups because the entire cohort included only middle-aged subjects with a mean age of 55 years and SD of 10 years. Therefore, the entire cohort did not reflect different age groups such as adolescents, middle-aged, and elderly subjects.
We hope this clarifies any ambiguities and confirms that we have considered the potential effect of age and sex in our analyses.
12. Discuss why no weight loss was observed despite LI.
We now provide a more detailed discussion on the absence of weight loss in the discussion in lines:201-207.
“In the studies conducted by Wasserfurth et el. and Dad'ová et al., subjects had a significantly lower BMI (28.4±5.8 kg/m² and 27.2±3.9 kg/m², respectively) compared to the present study (34.6±5.3 kg/m²). Among several other determinants, a higher initial BMI has been described as a barrier to behavioral change in LI [17]. In addition, the LI in Wasserfurth and Dad'ová et al differed from this study in that it was led by a professional trainer, took place regularly in a gym and was progressively increased, which may explain the greater effect on weight loss”.
13. Provide practical implications for dietary supplementation in obesity management.
We add more information in lines 50-58:
“Lifestyle interventions (LI), including dietary restrictions and increased physical activity, alongside pharmacotherapy, can result in clinically significant weight loss and enhanced insulin sensitivity [11]. Besides this basic therapy in obesity and T2DM management, dietary supplements with specific nutrients and bioactive compounds can contribute to overall metabolic and physiological functions. Notably, n3 PUFAs play a crucial role due to their anti-inflammatory properties and their ability to regulate glucose metabolism, lipid profiles, and insulin sensitivity, making them particularly beneficial for conditions such as metabolic syndrome or insulin resistance. However, it should be clearly stated that dietary supplements do not replace a balanced diet as well as physical exercise in managing obesity”.
14. Suggest longer follow-up studies or alternative insulin sensitivity assessments.
We have added the need for long-term follow-up studies as you suggest in the conclusion in lines 432-436: “Our data suggest that individuals with mild IR may benefit from CO supplementation, either at a high dose or at a lower dose in combination with LI, potentially leading to improved insulin sensitivity. The clinical relevance of these exploratory findings needs to be clarified through long-term follow-up studies”.
15. Discussion should include more clinical implications.
This is a good suggestion. We now discuss the following clinical implications:
- In lines 208-211 we note the importance of a personalized approach to identify preventive strategies: “In general, there is a debate about phenotyping the prediabetic population in order to improve prevention of progression to diabetes. Detailed, personalized information is essential for clinicians to better estimate the effectiveness of preventive strategies for different types of patients, leading to more targeted and effective treatment”
- In lines 238-246 we now discuss how varying doses and durations of administration may influence outcomes: “Even at the highest dose of CO (4 g CO/d: 276 mg EPA + 256 mg DHA), the amounts of n3 PUFAs are relatively low, compared to other studies in subjects with severe IR: A study by Derosa et al. observed a significant reduction in HOMA index compared to placebo in 281 overweight subjects with severe IR after 9 and 18 months of 3.0 g of n3 PUFA supplementation from fish oil in combination with a controlled energy diet. In contrast, the intervention in the CAGHO study was carried out without a controlled diet and for only 3 months, so we can only speculate whether higher doses of CO would also have significant effects in subjects with severe IR”.
- In lines 268-279 we now discuss the usefulness of additional markers of IR that would improve the clinical power of this study: “Liver fat content, which plays a key role in characterizing IR phenotypes and predicting improved insulin secretion, was not assessed before or after the intervention. We did not observe any changes in body weight, body fat content or calculated visceral fat mass, which is highly correlated with liver fat content. Therefore, the influence of changes in liv-er fat content is likely to be marginal. Insulin sensitivity was not assessed by the oral glucose tolerance test (oGTT) or the hyperinsulinemic-euglycemic clamp technique, which reflect more detailed methods of assessing glucose intolerance and beta-cell function, alt-hough alternative markers such as fasting glucose, HbA1c and the HOMA index were used instead. The use of the HOMA index is limited in subjects who do not have functioning beta cells. Additionally, the correlation between the HOMA index the hyperinsulinemic-euglycemic clamp technique is reduced in normal weight populations. However, this study used obese population without T2DM or T1DM subjects”.
- In lines 433 and 436 we now discuss the need for long-term follow-up observations as well as the possible combination of high dose CO with LI for more definitive clinical implications: “The clinical relevance of these exploratory findings needs to be clarified through long-term follow-up studies. Additionally, further research is needed to explore the impact of high doses of CO when combined with LI”.
References
[1] K.A. Abbott, T.L. Burrows, S. Acharya, R.N. Thota, M.L. Garg, DHA-enriched fish oil reduces insulin resistance in overweight and obese adults, Prostaglandins, leukotrienes, and essential fatty acids 159 (2020) 102154.
[2] G. Derosa, A.F.G. Cicero, A. D'Angelo, C. Borghi, P. Maffioli, Effects of n-3 pufas on fasting plasma glucose and insulin resistance in patients with impaired fasting glucose or impaired glucose tolerance, BioFactors (Oxford, England) 42 (2016) 316–322.
[3] W. Xia, N. Tang, H. Kord-Varkaneh, T.Y. Low, S.C. Tan, X. Wu et al., The effects of astaxanthin supplementation on obesity, blood pressure, CRP, glycemic biomarkers, and lipid profile: A meta-analysis of randomized controlled trials, Pharmacological research 161 (2020) 105113.
[4] J.-F. Wang, H.-M. Zhang, Y.-Y. Li, S. Xia, Y. Wei, L. Yang et al., A combination of omega-3 and plant sterols regulate glucose and lipid metabolism in individuals with impaired glucose regulation: a randomized and controlled clinical trial, Lipids in health and disease 18 (2019) 106.
[5] G. Marazzi, L. Cacciotti, F. Pelliccia, L. Iaia, M. Volterrani, G. Caminiti et al., Long-term effects of nutraceuticals (berberine, red yeast rice, policosanol) in elderly hypercholesterolemic patients, Advances in therapy 28 (2011) 1105–1113.
[6] T. Čížková, M. Štěpán, K. Daďová, B. Ondrůjová, L. Sontáková, E. Krauzová et al., Exercise Training Reduces Inflammation of Adipose Tissue in the Elderly: Cross-Sectional and Randomized Interventional Trial, The Journal of clinical endocrinology and metabolism 105 (2020).
[7] K. Daďová, M. Petr, M. Šteffl, L. Sontáková, M. Chlumský, M. Matouš et al., Effect of Calanus Oil Supplementation and 16 Week Exercise Program on Selected Fitness Parameters in Older Women, Nutrients 12 (2020).
[8] K.S. Tande, T.D. Vo, B.S. Lynch, Clinical safety evaluation of marine oil derived from Calanus finmarchicus, Regulatory toxicology and pharmacology RTP 80 (2016) 25–31.
[9] M. Burhop, J.P. Schuchardt, J. Nebl, M. Müller, R. Lichtinghagen, A. Hahn, Marine Oil from C. finmarchicus Enhances Glucose Homeostasis and Liver Insulin Resistance in Obese Prediabetic Individuals, Nutrients 14 (2022).
[10] P. Wasserfurth, J. Nebl, J.P. Schuchardt, M. Müller, T.K. Boßlau, K. Krüger et al., Effects of Exercise Combined with a Healthy Diet or Calanus finmarchicus Oil Supplementation on Body Composition and Metabolic Markers-A Pilot Study, Nutrients 12 (2020).
[11] M. Dempsey, M.S. Rockwell, L.M. Wentz, The influence of dietary and supplemental omega-3 fatty acids on the omega-3 index: A scoping review, Frontiers in nutrition 10 (2023) 1072653.
[12] International Society for the Study of Fatty Acids and Lipids (ISSFAL). PUFA Recommendations. Available online: http://www.issfal.org/statements/pufa-recommendations
[13] L.K. Oharomari, M.J. Ikemoto, D.J. Hwang, H. Koizumi, H. Soya, Benefits of Exercise and Astaxanthin Supplementation: Are There Additive or Synergistic Effects?, Antioxidants 10 (2021).
[14] L.-C. Tao, J.-N. Xu, T.-T. Wang, F. Hua, J.-J. Li, Triglyceride-glucose index as a marker in cardiovascular diseases: landscape and limitations, Cardiovascular diabetology 21 (2022) 68.
[15] E.S. Kang, Y.S. Yun, S.W. Park, H.J. Kim, C.W. Ahn, Y.D. Song et al., Limitation of the validity of the homeostasis model assessment as an index of insulin resistance in Korea, Metabolism: clinical and experimental 54 (2005) 206–211.
[16] K.-E. Eilertsen, H.K. Mæhre, I.J. Jensen, H. Devold, J.O. Olsen, R.K. Lie et al., A wax ester and astaxanthin-rich extract from the marine copepod Calanus finmarchicus attenuates atherogenesis in female apolipoprotein E-deficient mice, The Journal of nutrition 142 (2012) 508–512.
[17] E. Burgess, P. Hassmén, K.L. Pumpa, Determinants of adherence to lifestyle intervention in adults with obesity: a systematic review, Clinical obesity 7 (2017) 123–135.
[18] WHO Guidelines on Physical Activity and Sedentary Behaviour, 1st ed., World Health Organization, Geneva, 2020.
[19] M.D. DeBoer, S.L. Filipp, M.J. Gurka, Use of a Metabolic Syndrome Severity Z Score to Track Risk During Treatment of Prediabetes: An Analysis of the Diabetes Prevention Program, Diabetes care 41 (2018) 2421–2430.
[20] D.L. Tahapary, L.B. Pratisthita, N.A. Fitri, C. Marcella, S. Wafa, F. Kurniawan et al., Challenges in the diagnosis of insulin resistance: Focusing on the role of HOMA-IR and Tryglyceride/glucose index, Diabetes & metabolic syndrome 16 (2022) 102581.
Reviewer 3 Report
Comments and Suggestions for Authors
Lines 25-26: This conclusion is not clear. How may the effect of "CO supplementation with LI" vary by "combination with LI"?
The lifestyle intervention is not described. What was the lifestyle intervention? The data provided in the Methods section (lines 271-272) are far from sufficient.
Changes in physical activity before and after supplementation were assessed. From the discussion it appears that the authors expected changes. Why?
How was the “Metabolic Syndrome Severity (MetS) Score” calculated?
Figure 2 shows the estimated data. How do these data relate to the changes observed in the study group and presented in Table 2? The data presented in Table 2 indicate that the actual changes in the measured parameters were small and it is difficult to assess their clinical relevance. In how many cases were positive changes observed in the tested parameters, i.e. changes of at least 10% of the baseline value?
What is the clinical significance of statistical calculations regarding group*IR interactions? This aspect was omitted from both the discussion and the results. Statistical evaluations of experimental data are not valuable in themselves. Their purpose is to help us understand biological relationships and to evaluate observed changes and the possibilities of their use in practice, in preventive or therapeutic activities.
Key question: What is the biological/clinical relevance of the data presented?
The data shown in Figure 3 clearly indicate that the biological effect was observed only in the mild IR group and only when using 4g CO/d or 2g CO/d with LI.
HOMA is calculated based on insulin and glucose concentrations. Why changes in insulin and glucose concentrations are described as secondary outcomes? What are the primary outcomes?
HOMA is calculated on the basis of insulin and glucose concentrations. Why are changes in insulin and glucose levels called secondary outcomes? What are the primary outcomes?
In general, the conclusion is not supported by the results presented.
Author Response
1. Lines 25-26: This conclusion is not clear. How may the effect of "CO supplementation with LI" vary by "combination with LI?
The reviewer is right. This was not clearly stated. We not state: “CO supplementation improved glucose homeostasis, with effects varying by dose, combination with LI and IR phenotype”.
2. The lifestyle intervention is not described. What was the lifestyle intervention? The data provided in the Methods section (lines 271-272) are far from sufficient.
Thank you for your comment. We added more information on LI in “Section 4.4 Monitoring of lifestyle intervention and physical activity” (lines 336-344): “Subjects in the 2 g CO/d + LI group were advised to engage in regular exercise in accordance with WHO recommendations [18]. In detail, subjects were instructed to perform at least 150-300 minutes of moderate physical activity or at least 75-150 minutes of vigorous physical activity. To monitor LI adherence and weekly physical activity, subjects received a physical activity diary at t0 and t12. Those reporting <85% of the instructed activity per week, showing strong fluctuations between weeks, or achieving <85% overall were excluded from the analysis. Diet counselling included general recommendations from national professional societies for healthier eating”.
3. Changes in physical activity before and after supplementation were assessed. From the discussion it appears that the authors expected changes. Why?
Thanks for your comment. This was not clearly described. We now state in lines 350-353: “Subjects randomized in group of 2 g CO/d, 4 g CO/d, and placebo were instructed not to change their dietary habits (especially regarding the intake of n3 PUFA-rich foods) or physical activity during the intervention period to minimize dietary effects on variability in n3 PUFA status and glucose metabolism”.
Therefore, we expected changes in physical activity only in the 2 g CO/d + LI group.
4. How was the “Metabolic Syndrome Severity (MetS) Score” calculated?
Thanks for your advice on this. The calculation of the Metabolic Syndrome Severity Score is stated in lines 336-374: “The browser-based American Metabolic Syndrome (MetS) Severity Calculator (https://metscalc.org/metscalc/) was used to calculate MetS severity scores for each subject. The calculated MetS severity score was first described by Gurka and De Boer et al. [19] and takes into account the following cardiovascular disease risk parameters: systolic blood pressure, TG, HDL-C, fasting glucose, as well as information on sex, age, race/ethnicity, and weight. As a result, a single value based on BMI and a single value based on WC are calculated for each person”. The detailed formula cannot be provided because the calculation is done on a browser-based calculator with the publication rights of the authors Gurka and De Boer et al.
5. Figure 2 shows the estimated data. How do these data relate to the changes observed in the study group and presented in Table 2? The data presented in Table 2 indicate that the actual changes in the measured parameters were small and it is difficult to assess their clinical relevance. In how many cases were positive changes observed in the tested parameters, i.e. changes of at least 10% of the baseline value?
Thanks for your comment. Estimated changes reflect changes in the HOMA index relative to placebo. Figure 2 visualizes these estimated changes for all CO treated groups across different initial HOMA index values created from the statistical model described in lines 412-425 “Due to the exploratory nature of this study, a simple slope analysis (Figure 2) including main group effects, linear and squared interactions with HOMA index at t0 was used to identify areas of IR that were most responsive to CO supplementation. To determine the effects of CO supplementation, analysis of covariance (ANCOVA) were performed with change in HOMA index, insulin, glucose or HbA1c (Δt12- t0) as dependent variables. Study group was used as a fixed factor. Centered measurement of HOMA index; insulin; glucose and HbA1c before the intervention (t0); squared measurement of HOMA index; insulin; glucose; HbA1c at t0; BMI; age and sex were used as covariates. Squared interaction was entered because, based on simple slope analysis, a nonlinear moderator effect of initial HOMA index levels on the effect of treatment was suspected. Age and BMI were used as covariates because age and BMI were significantly associated with measurements of HOMA index, insulin, glucose and HbA1c at t0. If significant, a post hoc test was per-formed to determine the difference between the study groups”.
This means that Figure 2 considers the effects of CO treatment on each HOMA index value. In contrast, Table 2 shows the mean and standard deviation of HOMA index values before and after the intervention, aggregated over all initial HOMA index values. The effects presented in Table 2 are attenuated compared to those in Figure 2 because the effect within groups is smaller than in the placebo group, and the overall effects across the entire cohort are less pronounced than those shown in Figure 2, particularly in subjects with mild insulin resistance (IR).
Cases of reduction in HOMA index of more than 10% were as follows: 2 g CO/d group: n=21 (34%); 4 g CO/d group: n=27 (44%); 2 g CO/d + LI group: n=27 (47%) and placebo: n=24 (41%). The statistical model used in this study is based on a non-linear relationship between the initial HOMA index level and the change in HOMA index after CO treatment, with age, BMI and sex as covariates. The simple comparison of "responders" may lead to misinterpretation.
6. What is the clinical significance of statistical calculations regarding group*IR interactions? This aspect was omitted from both the discussion and the results. Statistical evaluations of experimental data are not valuable in themselves. Their purpose is to help us understand biological relationships and to evaluate observed changes and the possibilities of their use in practice, in preventive or therapeutic activities.
The reviewer is correct. This needs to be clarified. The clinical significance of the interaction term group* squared HOMA index values is the observation that the effect of CO supplementation alone or in combination with LI on the HOMA index followed a non-linear association. This means that in subjects with unlikely IR no effects could be seen, and in subjects with a severe IR the positive effects of CO are attenuated in the same manner. Only in subjects with mild IR the intervention was biological active. We now state in lines 218-225: “This finding is significant for clinical considerations, as it highlights the variability in treatment effects and underscores the importance of tailoring CO interventions to individual patient profiles, particularly in those with mild IR. Dietary supplements containing CO may be beneficial in a preventive approach when the progression of IR is at an early stage. In severe IR, CO had no further benefit, as no therapeutic approach can be expected from dietary supplements. In such metabolic disorders, LI, including dietary supplements, may be underpowered”.
7. Key question: What is the biological/clinical relevance of the data presented?
We have formulated this accordingly in the revised conclusion (lines 432-436): “Our data suggest that individuals with mild IR may benefit from CO supplementation, either at a high dose or at a lower dose in combination with LI, potentially leading to improved insulin sensitivity. The clinical relevance of these exploratory findings needs to be clarified through long-term follow-up studies.”
8. The data shown in Figure 3 clearly indicate that the biological effect was observed only in the mild IR group and only when using 4 g CO/d or 2 g CO/d with LI.
Thanks for your comment. In agreement with this in line 108-111 we state: “The effect of CO was significant only for 2 g CO/d + LI compared to placebo (estimated marginal means [EMM] 95% confidence interval [CI]: -0.19 [-0.80 - 0.41], p=0.021)”. In addition, in line 131-134 we state: “significant effect on HOMA index was among subjects with mild IR (Figure 3 b), with a difference in the change of HOMA index between the 4 g CO/d vs placebo group (EMM [95 % CI]: -0.76 [-1.53 – 0.03], p=0.043) and between the 2 g CO/d + LI vs placebo group (EMM [95 % CI]: -0.76 [-1.51 – 0.02], p=0.032)”.
In lines 432-436 we have added information on the biological/clinical relevance of these findings, as mentioned above.
9. HOMA is calculated based on insulin and glucose concentrations. Why changes in insulin and glucose concentrations are described as secondary outcomes? What are the primary outcomes?
Thank you for your comment. This was not clearly stated. We now clearly define in the method section lines 375-376 the primary outcome. In addition, we have added a brief explanation of the distinction between the HOMA index and fasting glucose and fasting insulin in lines 378-384: “The HOMA index was used as the primary outcome instead of fasting glucose and fasting insulin because the HOMA index measures IR by assuming feedback between the liver and beta cells. Glucose concentrations are regulated by insulin-dependent glucose production in the liver, while insulin levels depend on the response of pancreatic beta cells to glucose concentrations. Therefore, a reduced response to glucose-stimulated insulin secretion reflects a deficiency in beta cell function. IR can be observed by the reduced suppressive effect of insulin on glucose production in the liver” [20]. In addition, we will modify Table 2 to make it even easier to distinguish between primary and secondary outcomes.
The validity of the HOMA index is described in the introduction, we revised this part and now state in lines 39-46: “It allows assessment of intrinsic beta cell function and insulin sensitivity and has been vali-dated against the gold standard method for assessing IR (hyperinsulinemic euglycemic clamp), which can only be used in small studies. The HOMA index has been widely used to describe IR, but the thresholds that identify individuals at risk of developing T2DM may vary depending on the population, ethnicity, and health status. In the Jackson Heart Study, the HOMA index was used as a marker of IR in an obese population, offering additional stratification of T2DM risk beyond obesity alone.”
10. In general, the conclusion is not supported by the results presented.
The reviewer is right, we changed the conclusion and now state in lines 432-436: “Our data suggest that individuals with mild IR may benefit from CO supplementation, either at a high dose or at a lower dose in combination with LI, potentially leading to improved insulin sensitivity. The clinical relevance of these exploratory findings needs to be clarified through long-term follow-up studies”.
References
[1] K.A. Abbott, T.L. Burrows, S. Acharya, R.N. Thota, M.L. Garg, DHA-enriched fish oil reduces insulin resistance in overweight and obese adults, Prostaglandins, leukotrienes, and essential fatty acids 159 (2020) 102154.
[2] G. Derosa, A.F.G. Cicero, A. D'Angelo, C. Borghi, P. Maffioli, Effects of n-3 pufas on fasting plasma glucose and insulin resistance in patients with impaired fasting glucose or impaired glucose tolerance, BioFactors (Oxford, England) 42 (2016) 316–322.
[3] W. Xia, N. Tang, H. Kord-Varkaneh, T.Y. Low, S.C. Tan, X. Wu et al., The effects of astaxanthin supplementation on obesity, blood pressure, CRP, glycemic biomarkers, and lipid profile: A meta-analysis of randomized controlled trials, Pharmacological research 161 (2020) 105113.
[4] J.-F. Wang, H.-M. Zhang, Y.-Y. Li, S. Xia, Y. Wei, L. Yang et al., A combination of omega-3 and plant sterols regulate glucose and lipid metabolism in individuals with impaired glucose regulation: a randomized and controlled clinical trial, Lipids in health and disease 18 (2019) 106.
[5] G. Marazzi, L. Cacciotti, F. Pelliccia, L. Iaia, M. Volterrani, G. Caminiti et al., Long-term effects of nutraceuticals (berberine, red yeast rice, policosanol) in elderly hypercholesterolemic patients, Advances in therapy 28 (2011) 1105–1113.
[6] T. Čížková, M. Štěpán, K. Daďová, B. Ondrůjová, L. Sontáková, E. Krauzová et al., Exercise Training Reduces Inflammation of Adipose Tissue in the Elderly: Cross-Sectional and Randomized Interventional Trial, The Journal of clinical endocrinology and metabolism 105 (2020).
[7] K. Daďová, M. Petr, M. Šteffl, L. Sontáková, M. Chlumský, M. Matouš et al., Effect of Calanus Oil Supplementation and 16 Week Exercise Program on Selected Fitness Parameters in Older Women, Nutrients 12 (2020).
[8] K.S. Tande, T.D. Vo, B.S. Lynch, Clinical safety evaluation of marine oil derived from Calanus finmarchicus, Regulatory toxicology and pharmacology RTP 80 (2016) 25–31.
[9] M. Burhop, J.P. Schuchardt, J. Nebl, M. Müller, R. Lichtinghagen, A. Hahn, Marine Oil from C. finmarchicus Enhances Glucose Homeostasis and Liver Insulin Resistance in Obese Prediabetic Individuals, Nutrients 14 (2022).
[10] P. Wasserfurth, J. Nebl, J.P. Schuchardt, M. Müller, T.K. Boßlau, K. Krüger et al., Effects of Exercise Combined with a Healthy Diet or Calanus finmarchicus Oil Supplementation on Body Composition and Metabolic Markers-A Pilot Study, Nutrients 12 (2020).
[11] M. Dempsey, M.S. Rockwell, L.M. Wentz, The influence of dietary and supplemental omega-3 fatty acids on the omega-3 index: A scoping review, Frontiers in nutrition 10 (2023) 1072653.
[12] International Society for the Study of Fatty Acids and Lipids (ISSFAL). PUFA Recommendations. Available online: http://www.issfal.org/statements/pufa-recommendations
[13] L.K. Oharomari, M.J. Ikemoto, D.J. Hwang, H. Koizumi, H. Soya, Benefits of Exercise and Astaxanthin Supplementation: Are There Additive or Synergistic Effects?, Antioxidants 10 (2021).
[14] L.-C. Tao, J.-N. Xu, T.-T. Wang, F. Hua, J.-J. Li, Triglyceride-glucose index as a marker in cardiovascular diseases: landscape and limitations, Cardiovascular diabetology 21 (2022) 68.
[15] E.S. Kang, Y.S. Yun, S.W. Park, H.J. Kim, C.W. Ahn, Y.D. Song et al., Limitation of the validity of the homeostasis model assessment as an index of insulin resistance in Korea, Metabolism: clinical and experimental 54 (2005) 206–211.
[16] K.-E. Eilertsen, H.K. Mæhre, I.J. Jensen, H. Devold, J.O. Olsen, R.K. Lie et al., A wax ester and astaxanthin-rich extract from the marine copepod Calanus finmarchicus attenuates atherogenesis in female apolipoprotein E-deficient mice, The Journal of nutrition 142 (2012) 508–512.
[17] E. Burgess, P. Hassmén, K.L. Pumpa, Determinants of adherence to lifestyle intervention in adults with obesity: a systematic review, Clinical obesity 7 (2017) 123–135.
[18] WHO Guidelines on Physical Activity and Sedentary Behaviour, 1st ed., World Health Organization, Geneva, 2020.
[19] M.D. DeBoer, S.L. Filipp, M.J. Gurka, Use of a Metabolic Syndrome Severity Z Score to Track Risk During Treatment of Prediabetes: An Analysis of the Diabetes Prevention Program, Diabetes care 41 (2018) 2421–2430.
[20] D.L. Tahapary, L.B. Pratisthita, N.A. Fitri, C. Marcella, S. Wafa, F. Kurniawan et al., Challenges in the diagnosis of insulin resistance: Focusing on the role of HOMA-IR and Tryglyceride/glucose index, Diabetes & metabolic syndrome 16 (2022) 102581.
Reviewer 4 Report
Comments and Suggestions for Authors The article is very interesting, relevant, important and is devoted to assessing the effect of taking different doses of Calanus oil, including with lifestyle changes, on carbohydrate metabolism and metabolic syndrome. The study design is very competent, this is a clinical trial. The study is randomized, double-blind, placebo-controlled. Important results of the study were obtained. The authors concluded that 12 weeks of taking Calanus oil and positive lifestyle changes had a beneficial effect on glucose homeostasis. The effects varied depending on the progression of insulin resistance, the doses of oil administered and their combination with lifestyle changes. The authors themselves described many limitations of the study and, therefore, very correctly and carefully made a conclusion in the article. I have a small remark: 1. In the "Material and Methods" section, it is indicated that pregnant women were excluded from the study. It is also indicated that there were more than 66% women, that the age was from 30 to 75 years. So some of the women were before menopause, and some were after. Was this fact assessed during randomization into 4 groups of patients, was the level of female sex hormones in the blood monitored? This should be indicated and the "Material and Methods" section should be corrected.
Author Response
The article is very interesting, relevant, important and is devoted to assessing the effect of taking different doses of Calanus oil, including with lifestyle changes, on carbohydrate metabolism and metabolic syndrome. The study design is very competent, this is a clinical trial. The study is randomized, double-blind, placebo-controlled. Important results of the study were obtained. The authors concluded that 12 weeks of taking Calanus oil and positive lifestyle changes had a beneficial effect on glucose homeostasis. The effects varied depending on the progression of insulin resistance, the doses of oil administered and their combination with lifestyle changes. The authors themselves described many limitations of the study and, therefore, very correctly and carefully made a conclusion in the article. I have a small remark: 1. In the "Material and Methods" section, it is indicated that pregnant women were excluded from the study. It is also indicated that there were more than 66% women, that the age was from 30 to 75 years. So some of the women were before menopause, and some were after. Was this fact assessed during randomization into 4 groups of patients, was the level of female sex hormones in the blood monitored? This should be indicated and the "Material and Methods" section should be corrected.
We kindly thanks for your comment. We did not evaluate female sex hormones. However, we randomized according to age, sex, BMI, and WC. Under these conditions, a different distribution of pre- and postmenopausal women would not be expected. In consequence Table 1 show that subjects within the groups did not differ in mean age or in sex distribution. This means that pre- and postmenopausal women are equally divided between the groups.
Round 2
Reviewer 1 Report
Comments and Suggestions for Authors
The manuscript is improved after the revision.
Author Response
We kinfly thanks for acception
Reviewer 2 Report
Comments and Suggestions for Authors
Accept.
Author Response
We kindly thanks for acception.
Reviewer 3 Report
Comments and Suggestions for Authors
The manuscript presents the results of interesting research. The current version meets the requirements of scientific journals. Only minor changes are required.
Table 2 and results description: I suggest removing the "primary" and "secondary" endpoints because they are misleading to the reader and do not add anything to the evaluation of the results. We measure the glucose and insulin concentrations from which we calculate HOMA. HOMA is not evaluated independently of the observed glucose and insulin concentrations.
Author Response
Table 2 and results description: I suggest removing the "primary" and "secondary" endpoints because they are misleading to the reader and do not add anything to the evaluation of the results. We measure the glucose and insulin concentrations from which we calculate HOMA. HOMA is not evaluated independently of the observed glucose and insulin concentrations.
Thank you for your comment. We follow your suggestion and remove the primary and secondary endpoints from Table 2. We also remove data of fasting insulin and fasting glucose concentration from Table 2. These data are now reported in Table S5 in the Supplementary part.